# Multi-Omic Profiles in Infants at Risk for Food Reactions

**DOI:** 10.3390/genes13112024

**Published:** 2022-11-03

**Authors:** Ramin Beheshti, Shane Stone, Desirae Chandran, Steven D. Hicks

**Affiliations:** 1Penn State Health Milton S Hershey Medical Center, Department of Pediatrics, Hershey, PA 17033, USA; 2Department of Pediatrics, Penn State Health Children’s Hospital, 500 University Drive, Hershey, PA 17033, USA

**Keywords:** food reactions, atopy, cytokines, microbiome, food allergy

## Abstract

Food reactions (FR) are multifactorial and impacted by medical, demographic, environmental, and immunologic factors. We hypothesized that multi-omic analyses of host-microbial factors in saliva would enhance our understanding of FR development. This longitudinal cohort study included 164 infants followed from birth through two years. The infants were identified as FR (*n* = 34) or non-FR (*n* = 130) using the Infant Feeding Practice II survey and medical record confirmation. Saliva was collected at six months for the multi-omic assessment of cytokines, mRNAs, microRNAs, and the microbiome/virome. The levels of one miRNA (miR-203b-3p, adj. *p* = 0.043, V = 2913) and one viral phage (Proteus virus PM135, adj. *p* = 0.027, V = 2955) were lower among infants that developed FRs. The levels of one bacterial phylum (Cyanobacteria, adj. *p* = 0.048, V = 1515) were higher among infants that developed FR. Logistical regression models revealed that the addition of multi-omic features (miR-203b-3p, Cyanobacteria, and Proteus virus PM135) improved predictiveness for future FRs in infants (*p* = 0.005, X^2^ = 12.9), predicting FRs with 72% accuracy (AUC = 0.81, sensitivity = 72%, specificity = 72%). The multi-omic analysis of saliva may enhance the accurate identification of infants at risk of FRs and provide insights into the host/microbiome interactions that predispose certain infants to FRs.

## 1. Introduction

Food reactions (FR) are adverse responses that arise from immune and non-immune (food intolerance) processes and can vary in presentation and pathophysiological origin [1]. Immune-mediated FRs can be Immunoglobulin E (IgE)-mediated or non-IgE-mediated [1]. Food allergies (FA) are IgE-mediated FRs that continue to be a major health concern worldwide [2]. Most cases of FRs develop in the first year of life, suggesting that the infancy period is a crucial time for the development of FRs [3]. The onset of FRs is thought to be caused by an interplay of environmental, genetic, and epigenetic factors [3].

Several environmental factors have been associated with risk for FRs [4]. This includes the timing of food introduction, the delivery method, breast versus formula feeding, tobacco exposure, the mechanism of delivery, prematurity, low birth weight, and antibiotic exposures [4]. Studies have demonstrated the prenatal period as a critical time for immune programming and linked maternal environmental exposures during pregnancy to increased infantile FR risk [5]. A meta-analysis study demonstrated that Cesarean delivery was associated with an increased risk of FA-sensitive phenotypes [6]. The hygiene hypothesis suggests that microbial colonization in the infancy period may influence FA risk [7]. Environmental influences such as a lack of breastfeeding, antibiotics, and exposure to pollution and farm animals may also impact the infantile microbiome, modulating immune development and FR risk [7].

Studies have also determined that delayed food introduction increases the risk the for development of FAs [8]. The Learning Early About Peanut (LEAP) trial demonstrated that, in infants with severe eczema and/or egg allergy (classified as high-risk infants), the early introduction of peanuts between 4 and 11 months of age resulted in a reduction in the development of peanut allergies [9]. The identification of allergen subcomponents could lead to a better understanding of the allergic protection associated with early introduction. Reliable methodologies for the protein and DNA detection of allergens have recently been developed for common antigens such as tree nuts [10]. DNA-based methodologies, including real-time PCR, microarrays, and DNA biosensors, may provide a more reliable alternative to standard enzyme-linked immunosorbent assays [10].

Genetic–environmental interactions also have an impact on the risk of FRs [11,12]. Infants who develop FAs by the toddler stage have higher TH2 cytokine responses, suggesting that allergen-specific immunologic responses in the first year of life correlate with the risk of the development of FAs [3,12]. Several gene polymorphisms have been linked to FAs, and most of these genes appear to be involved in antigen presentation or a shift towards a TH2 response [3,12]. It is likely that interactions between genetic and environmental factors contribute to FRs [13].

Early life environmental exposures and gene–environment interactions are mediated by epigenetic alterations, which impact the development of FRs [14,15]. Epigenetic modifications are heritable changes in the expression of non-coding genes [15]. Three examples have been implicated in modulating gene expression, including DNA methylation, histone modification, and the expression of microRNAs (miRNAs) [15]. The latter is responsible for post-transcriptionally regulating the expression of a network of target genes [14,15]. This is particularly important in the development of FRs since miRNAs can respond to environmental stimuli by altering the expression of genes implicated in FRs [14,15]. The plasticity of the epigenome suggests variation in early life environmental exposure can profoundly alter phenotypes through epigenetic effects, explaining why genetic predisposition alone may not account for the individual risk of FRs [15].

Gene environment interactions may also be influenced by microbial composition [16]. Vitamin D serves as an immunological regulator of microbiome dysbiosis, which influences the pro-inflammatory processes in atopic diseases [16,17]. A maternal deficiency of vitamin D has been linked to the development of food allergies in infancy [16,17]. Additionally, microbial composition regulates immune function, and the preservation of the floral microbiome may prevent the onset of allergic disease [16]. The role of probiotics in the preservation of the “natural” microbiome is not fully understood, but studies have suggested that the maternal supplementation of vitamin D and probiotics during pregnancy may modulate protection from food allergy development [16].

Although various environmental exposures, genetics, and epigenetic factors have been implicated in FR pathophysiology, to our knowledge, few studies have employed a multi-omic approach to explore their interactions. Such information could provide insight into the pathophysiologic mechanisms underlying FRs, leading to the development of novel diagnostic, prevention, and treatment approaches. Additionally, clinicians have few tools to predict the risk for FR development in healthy infants. We hypothesized that interactions between specific microbes, miRNAs, mRNAs, and cytokines modulate FR development, and may be used to predict FR risk. This hypothesis was tested using a longitudinal cohort study of 164 infants from birth through two years of age.

## 2. Methods

### 2.1. Participants

This prospective longitudinal cohort comprised a convenience sample of 221 infants. The eligibility criteria were term delivery (37–42 weeks gestation) and the intention to breastfeed for at least four months. The exclusion criteria were maternal conditions that could impact breastfeeding ability (e.g., drug addiction, HIV), infant conditions that could impact feeding (e.g., cleft lip/palate, metabolic disease, prolonged NICU admission >7 days), factors impacting longitudinal follow-up (e.g., plan for primary pediatric care outside our medical center, plan for infant adoption), and non-English speaking. Infants were enrolled within seven days of delivery. Enrollment occurred between 20 April 2018 and 5 October 2020 at the newborn nursery or the outpatient pediatrics clinics affiliated with our academic medical center. We screened 2487 infants for eligibility, of whom 359 met the criteria, 221 consented to participate, and 164 completed sufficient follow-up surveys for this study. Infants who completed the study were dichotomized into two groups based on the presence or absence of FRs in the first two years. An FR was determined through the parents reporting food sensitivity, intolerance, or allergy reaction on the Infant Feeding Practices (IFP)-II survey (administered at 1, 4, 6, and 12 months), and the documentation of food allergy in the electronic medical record [18]. The final analysis included 130 infants without FRs and 34 infants with FRs (Figure 1).

### 2.2. Measures

Electronic surveys administered by research staff were used to obtain medical and demographic characteristics from all infants at enrollment. The following medical and demographic characteristics were collected: maternal age (years), maternal body mass index (BMI; kg/m^2^), maternal history of food allergies, maternal history of asthma and allergic rhinitis, family history of food allergies in a first- or second-degree relative, infant biological sex, infant gestational age (weeks), infant race, and infant birth weight (g). The National Survey of Lead and Allergens in Housing (NSLAH) was used to assess the following environmental factors that could impact allergy outcomes: delivery mode (vaginal, cesarean section), maternal tobacco use, earliest formula introduction (<4 weeks, 4–16 weeks, 16–24 weeks, or >24 weeks), the timing of solid food introduction, and the number of people living in the household [19]. The IFP was administered at 1, 4, 6, and 12 months to assess for infant dietary habits, including the timing and type of solid food intake [18]. A review of the medical record was used to confirm the timing, character (anaphylactic or non-anaphylactic), and inciting food for each FR.

### 2.3. Sample Collection and Processing

Two non-fasting saliva samples were collected from each infant at six months. One swab contained a nucleic acid stabilizer for RNA analysis (DNA Genotek, Ottowa, ON, Canada). The second swab contained no stabilizing reagents (Oasis Diagnostics, Vancouver, WA, Canada) and was used for cytokine analysis. The samples were collected from the sublingual and parotid regions by placing each swab into the mouth for approximately 20 s. Samples were aliquoted and stored at −80 °C. The time of sample collection was recorded to ensure that circadian variation in molecular profiles did not contribute to between-group differences.

### 2.4. Cytokine Analysis

Concentrations of four cytokines were measured in infant saliva using enzyme-linked immunosorbent assays (ELISAs). Levels of C-X-C Motif Chemokine Ligand 10 (CXCL10), interleukin 18 (IL-18), interleukin 6 (IL-6), and interleukin 8 (IL-8) were chosen for assessment based on their nasopharyngeal bioavailability and their potential importance in allergic conditions [20]. Thawed saliva samples were centrifuged at 16,000× *g* for 4 min. Next, 70 µL of saliva was added to 70 µL of diluent and vortexed for 30 s. Then, the diluted solution was centrifuged at 1000× *g* for 2 min before adding an additional 50 µL of diluent. The diluted samples were pipetted onto a custom ELISA cartridge (SPCKC-PS-003661) and analyzed on an ELLA Simplex instrument (Protein Simple, San Jose, CA, USA).

### 2.5. RNA Analysis

RNA was extracted from each saliva sample using the miRNeasy Kit (Qiagen, Inc., Germantown, MD, USA). The integrity and quantity of each sample were measured using an Agilent Bioanalyzer 2100 (Agilent, Santa Clara, CA, USA). RNA libraries were prepared with the Illumina TruSeq Small RNA Prep protocol. High throughput sequencing was performed at the State University of New York Upstate Molecular Analysis Core with a NextSeq500 (New York City, NY, USA) instrument at a targeted depth of ten million, 50 base, paired-end reads per sample. The reads were aligned to the hg38 build of the human genome using Partek Flow (Partek; St. Louis, MO, USA) and the Bowtie2 aligner. The quantification of messenger RNAs (mRNAs) was performed using Ensembl annotation (v95), and the quantification of mature microRNAs was performed using miRBase 22 annotation. RNA reads that were unaligned to the human genome were aligned to the NCBI RefSeq genome using Kraken (v1). Bacteria were aligned at the phylum level and viral phages were aligned at the species level. We elected to measure microbial RNA (in lieu of a 16 S approach) to streamline our nucleic acid extraction and analyses protocol. The integrity of the RNA sequencing results was verified through read quality score and total read count. The RNA features with consistent detection (raw read counts ≥10 in 100% of samples) in each category (mRNAs, miRNAs, bacteria, virus) were sum-normalized and mean-center scaled prior to statistical analysis. Downstream analysis involved a priori interrogation of five mRNAs (STAT6, SPINK5, FLG, SERPINB2, MALT1) previously implicated in atopic conditions, as well as the 10 most abundant features within the miRNA, bacterial, and viral phage categories [21,22,23,24].

### 2.6. Statistical Analysis

Medical and demographic characteristics were compared between infants with and without FR using Student’s t-test or a Chi-squared test, as appropriate. A Wilcoxon Rank Sum Test was used to compare saliva cytokine levels, mRNAs, miRNAs, and microbial RNAs between the groups. Benjamini Hochberg false detection rate-adjusted *p*-values were reported. The multi-omic features that displayed differences among infants with and without FR (adjusted *p* < 0.05) were assessed for cross-category associations with Spearman’s rank correlation test. Hierarchical logistic regression was used to assess the ability of multi-omic features to identify infants at risk for FR, relative to medical and demographic risk factors. An initial model employing maternal/infant traits (maternal age, infant race, maternal history of FA) was compared against a second model using environmental exposures (delivery mode, earliest formula introduction, the timing of solid food introduction) and a third model, which added salivary levels of multi-omic features. The ability of each model to account for between-group variance was determined (McFadden’s R^2^), and significant gains across the models (*p* < 0.05) were reported. The predictive accuracy of each model was assessed by measuring the cross-validated area under a receiver operator characteristic curve (AUC), sensitivity, and specificity.

## 3. Results

### 3.1. Participants

The majority of infants were female (96/164; 58.5%) and White (126/164; 76.8%) (Table 1). Their average birth weight was 3352 g (±447) and their average gestational age was 38.9 (±1) weeks. The mothers of the participating infants had an average age of 30 (±4) years and an average BMI of 27.5 (±6) kg/m^2^. Few mothers had FAs (14/164; 8.5%), asthma (23/164, 14.0%), or allergic rhinitis (36/164, 21.9%). The majority of infants were delivered vaginally (132/164, 80.5%). Nearly half of the mothers reported no formula introduction in the first 6 months (67/153, 43.8%), and few introduced solid foods by 4 months (22/164, 13.4%). The average household had four occupants (range: 2–9 occupants). The mothers of infants with FRs were more likely to have a personal history of FAs (8/34, 23.5%; *p* = 0.00038, X^2^ = 12.6) than mothers of infants without FRs (6/130, 4%). There was no difference between the groups in other medical or demographic characteristics. 

### 3.2. Food Reaction Characteristics

The average age of the first FR was 8 (± 7) months. The majority of the infants (20/37, 54%) experienced their first FR after providing a saliva sample. The most common inciting foods were milk (12/34, 35.2%), egg (6/34, 17.6%), nuts (6/34, 17.6%), and fruits/vegetables (4/34, 11.7%). Approximately one-fifth (7/34, 20.5%) of participants developed anaphylactic symptoms and received a prescription for epinephrine.

### 3.3. Saliva Molecular Profiles

There was no difference in the average time of saliva collection between the groups (*p* > 0.05). None of the salivary cytokines (CXCL10, IL-8, IL-6, IL-18) revealed differences between the groups (Appendix A). The levels of one miRNA (miR-203b-3p, adj. *p* = 0.043, V = 2913) and one viral phage (*Proteus virus PM135*, adj. *p* = 0.027, V = 2955) were lower among infants that developed FRs (Figure 2). The levels of one bacterial phylum (*Cyanobacteria*, adj. *p* = 0.048, V = 1515) were higher among infants that developed FRs. None of the five mRNA transcripts displayed a difference between the groups (Appendix A).

### 3.4. Relationships between Molecular Factors Implicated in Food Reactions

All three of the omic features that differed between the children with and without FRs displayed cross-category relationships on Spearman Rank testing (Figure 3). The levels of miR-203b-3p were directly associated with *Cyanobacteria* levels (R = 0.53, *p* = 3.3 × 10^−13^), and inversely associated with *P*. *PM135* (R = −0.38, *p* = 5.2 × 10^−7^). The levels of *Cyanobacteria* and *P*. *PM135* were inversely associated (R = −0.39, *p* = 2.8 × 10^−7^).

### 3.5. Predicting Food Reactions

A logistic regression model employing maternal and infant traits (maternal age, infant race, maternal FA) accounted for 8.5% of the variance between the groups (*p* = 0.038) and predicted infant FR with 63% accuracy (AUC = 0.658, sensitivity = 59%, specificity = 64%). The addition of three environmental exposures (the delivery mode, the timing of formula introduction, and the introduction of solid foods at 4 months) accounted for an additional 5.1% of the variance between groups but did not significantly improve the model (*p* = 0.15, X^2^ = 8.0). However, the measurement of the three multi-omic features (miR-203b-3p, *Cyanobacteria*, and *P. PM135*) accounted for an additional 7.5% of the variance and significantly improved the model (*p* = 0.005, X^2^ = 12.9). The multi-omic features predicted FR with 72% accuracy (AUC = 0.81, sensitivity = 72%, specificity = 72%) when combined with medical and demographic traits. Maternal FA (*p* = 0.20, X^2^ = 5.40), miR-203b-3p (*p* = 0.032, X^2^ = 4.60), and Proteus virus (*p* = 0.019, X^2^ = 5.47) contributed most significantly to the predictive model. 

## 4. Discussion

In this cohort of 164 infants, we identified one trait (maternal history of FAs), one miRNA (miR-203b-3p), one microbe (*Cyanobacteria*), and one viral phage (*P. PM135*) that displayed potential differences between infants who developed FRs in the first two years, and infants who did not. Although medical and demographic traits provided some predictive value for identifying infants who developed FRs, the addition of three multi-omic features (miR-203b-3p, *Cyanobacteria*, and *P. PM135*) significantly enhanced FR prediction.

Factors such as maternal FAs, the timing of food introduction, and a history of atopic dermatitis have been correlated with FR risk, but to our knowledge, there are few clinical tools to identify healthy infants at risk for developing FRs [4]. Although the LEAP trial provided some insight into the role of early food exposure and FA protection, the application of this knowledge into a clinical risk stratification tool has not been validated [9]. Worldwide, FAs continue to burden patients and families, with recent models suggesting a significant increase in prevalence over the coming decades [21]. Anaphylaxis, as a result of FAs, remains a significant concern for parents of infants and children with FRs [21]. The ability to prospectively identify infants at risk for developing FRs would provide clinicians with an opportunity to deliver personalized anticipatory guidance. This could include mitigation strategies to ensure safe and timely exposure to allergenic foods or ensure that high-risk patients have access to epinephrine autoinjectors prior to the first episode of anaphylaxis. If a sufficiently specific predictive tool were developed, the results might be used to guide early referral, testing (i.e., skin prick test or radioallergosorbent test), or treatment (i.e., oral challenge plans). If successful, such a practice might decrease emergency visits and the need for hospitalizations due to severe anaphylaxis.

The results of this study provide insights into pathophysiologic mechanisms that may underlie infant FRs. Some “omic” features identified in this study have been described in previous studies of allergies. A study using canine models found miR-203 to be perturbed in atopic dermatitis when compared to healthy controls [25]. miR-203 is expressed in keratinocytes and the epithelial lining of the gastrointestinal tract [26] and has been associated with the dysregulation of tissue-specific gene targets, including *suppressor of cytokine signaling* (*SOCS3*) [27]. One study demonstrated that miR-203 signaling mediates the activation of anti-inflammatory pathways through *SOCS3* activity and the inhibition of the Janus kinase/signal transducer and activator of transcription proteins (JAK-STAT) pathway [28]. Notably, IL-6, a putative inflammatory mediator in atopy, is also a potent activator of the JAK-STAT pathway [29,30,31], which has been implicated in atopy and airway hyper-reactivity [32,33]. The results of the current study support the protective role of miR-203b in FRs and demonstrate that it can be non-invasively measured in infant saliva. Although we did not detect differences in salivary IL-6 in infants who developed FRs, this may have occurred because the samples were not taken at the time of the FR symptoms.

This study also identifies novel biological mechanisms that may underlie FR development. The oral mucosa maintains a diverse biogeographical continuum, where microbiota signatures may regulate host inflammation or reflect immunologic development [34]. For example, elevated levels of *Streptococcus* in saliva have been associated with perturbations in host cytokines that are linked to the activation of T helper 2 (Th2) cells and childhood food allergies [35]. *Cyanobacteria* abundance has been linked to neurodegenerative, gastrointestinal, hepatic, metabolic, and respiratory disease, although this mechanistic relationship is poorly understood [36]. In this study, the levels of *Cyanobacteria* were higher in infants who developed FRs, suggesting that *Cyanobacteria* abundance may mediate pro-inflammatory patterns associated with FRs. Additionally, the levels of *Cyanobacteria* were directly correlated with the expression of miR-203b-3p, suggesting that host epi-transcriptional machinery may modulate key microbial populations in individuals with FRs. Bacteriophages such as *P. PM135* are also active regulators of the microbiome [37]. Microbial dysregulation through alterations in phages abundance has been hypothesized in FA pathophysiology [38]. In this study, the levels of *P. PM135* were lower in the saliva of infants at risk for FRs. Thus, reductions in *P. PM135* may perturb the balance between various Proteus family members and promote the growth of specific bacterial populations that increase the risk for FRs. For example, the levels of *P. PM135* were inversely correlated with the levels of *Cyanobacteria* in this cohort. 

Based on these findings, we propose a conceptual framework for FR pathogenesis involving microbial perturbations, miR-203b-3p expression, and immune regulation (Figure 4). miR-203 has been found to decrease inflammation in tissue-specific targets through the activation of *SOCS3* [39], and this suppresses Th2 cytokine production (i.e., IL-4, IL-5, IL-6) through the JAK-STAT pathway [40,41]. Therefore, the decreased expression of miR-203b-3p, observed in children who developed FRs, may elicit a pro-inflammatory state. The secretion of IL-6 in the upper gastrointestinal tract may also be related to perturbations in microbial diversity [42]. In a study evaluating periodontitis, *Cyanobacteria* was demonstrated to modulate a pro-inflammatory response in the oral mucosa through a complex interaction of lipopolysaccharide (LPS)-mediated Toll-Like Receptor 4 (TLR4) blockade, IL-6 production, and miRNA expression [43]. Bacteriophages such as *P. PM135* secrete pathogen-associated molecular patterns (PAMPs) which can activate the host’s innate immune response and disrupt microbial colonization in the upper gastrointestinal tract [44]. Such disruption may serve to promote antimicrobial effects against “atopy favoring” colonization, protecting host susceptibility in developing pro-inflammatory FR.

There are several limitations that should be noted. FRs were identified through repeat validated surveys in addition to medical record reviews at two years of age. It is possible that some of the infants may have obtained medical attention for FRs at an outside facility and without parental reports. Participants were, however, followed at our outpatient pediatric clinic for routine well-child check-ups, at which time epinephrine autoinjector use and recent urgent care visits are typically recorded in the medical record. Samples of the saliva were collected at six months of age (prior to the first FR episode for over half of the participants), but the molecular perturbations identified here may not precede FR onset in all infants.

## 5. Conclusions

The results of this study support the premise that the multi-omic analysis of infant saliva may provide insight into pathophysiologic mechanisms that underlie food reactions, especially interactions between host immunity, microbial activity, and epitranscriptomic. The measurement of these multi-omic factors may also aid the identification of infants at risk for future food reactions. The ability to prospectively identify infants at risk for food reactions would allow clinicians to provide personalized guidance regarding the avoidance of allergenic foods and ensure that high-risk patients have access to immediate treatment such as epinephrine autoinjectors. If a sufficiently accurate tool were developed, the results might be used to guide early referral and treatment, which could one day decrease emergency visits and the need for hospitalizations due to severe food reactions.

## Figures and Tables

**Figure 1 genes-13-02024-f001:**
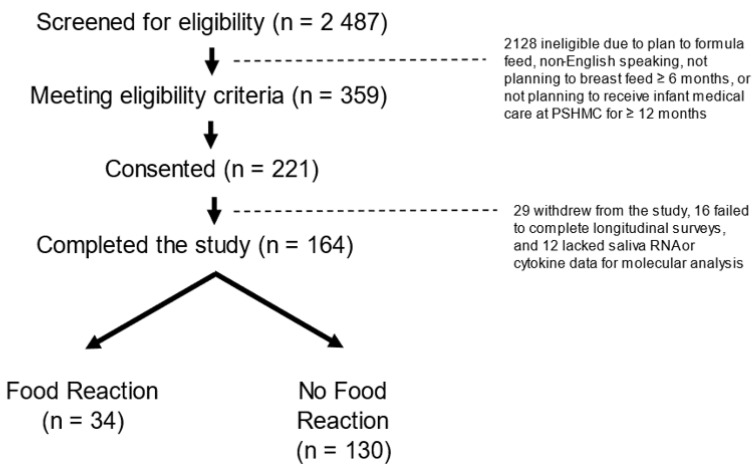
CONSORT Diagram: There were 2487 mother infant dyads screened, 359 eligible dyads approached, and 221 dyads consented. There were 164 infants for whom food allergy status could be confirmed in the first two years. There were 34 infants with food reactions (defined as parent reports of food sensitivity, intolerance, or allergy on the Infant Feeding Practice-II survey, and confirmed through a review of the medical record) and 130 infants without food reactions.

**Figure 2 genes-13-02024-f002:**
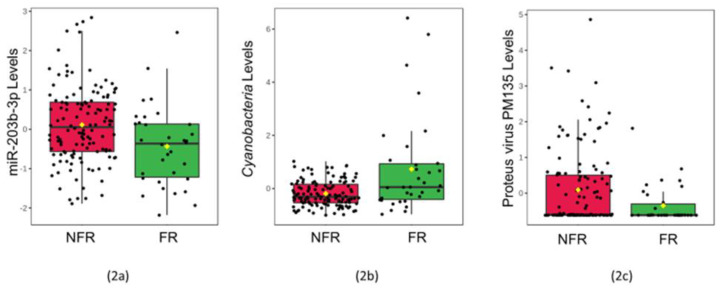
Saliva levels of three multi-omic features differ among infants with food reactions: The box plots display normalized concentrations of one miRNA (miR-203b-3p, adj. *p* = 0.043, V = 2913) (2**a**), one bacterial phylum (Cyanobacteria, adj. *p* = 0.048, V = 1515) (2**b**), and one viral species (Proteus virus PM135, adj. *p* = 0.027, V = 2955) (2**c**) that displayed significant differences (adj. *p* < 0.05) between 34 infants with food reactions (FR; green) and 130 peers with no food reactions (NFR; red) on Wilcoxon Rank Sum Testing. Normalized, scaled levels of each molecular feature are shown in the box plots, with the mean (black horizontal bar), median (yellow diamond), and standard error displayed.

**Figure 3 genes-13-02024-f003:**
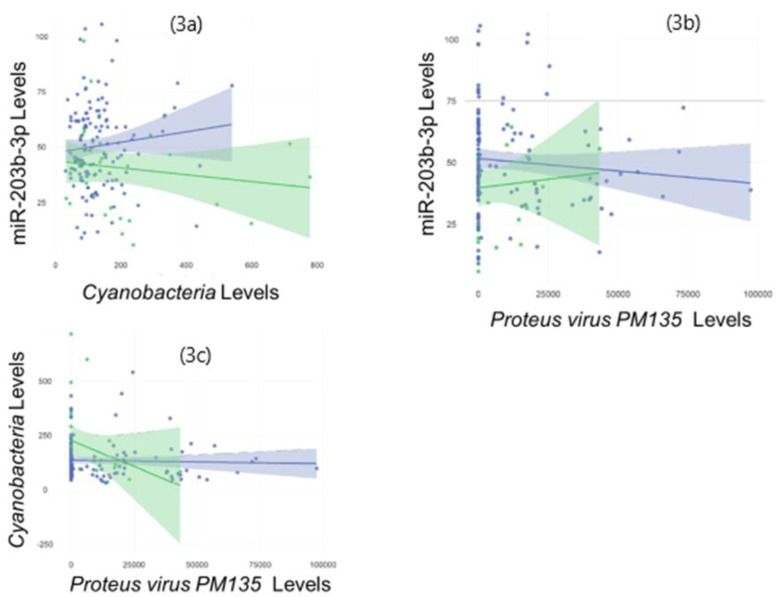
Multi-omic features implicated in food reactions display cross-category associations: The scatter plots display significant associations (*p* < 0.05) between all three molecular features of interest on Spearman Rank Testing. The levels of miR-203b-3p were directly associated with Cyanobacteria levels (R = 0.53, *p* = 3.3 × 10^−13^) (3**a**), and inversely associated with Proteus virus PM135 (R = −0.38, *p* = 5.2 × 10^−7^) (3**b**). The levels of Cyanobacteria and Proteobacteria virus PM135 were inversely associated (R = −0.39, *p* = 2.8 × 10^−7^) (3**c**). Separate trend lines with 95% confidence intervals are displayed for participants with food reactions (green) and no food reactions (blue).

**Figure 4 genes-13-02024-f004:**
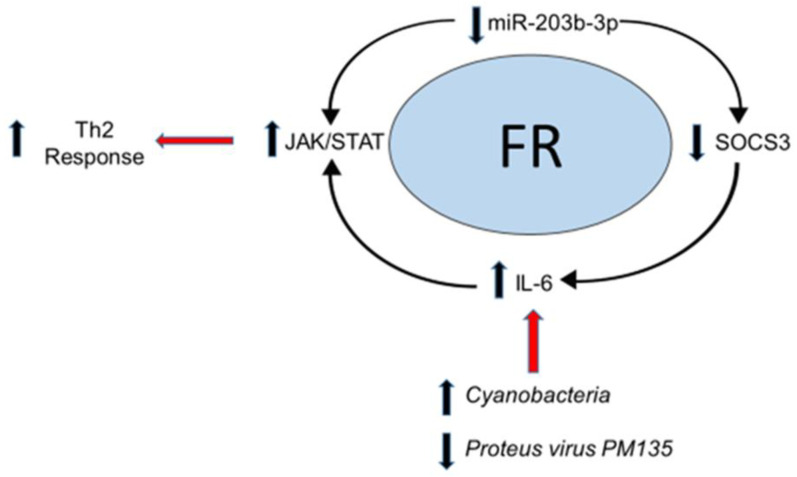
Conceptual framework model of host–microbial interactions in food reactions: The results of this study support a conceptual framework for FR pathophysiology involving host–microbiome interactions. Elevated levels of pathogenic Cyanobacteria may directly induce inflammatory cytokines such as interleukin-6 (IL-6). IL-6 is a known activator of the JAK-STAT pathway, resulting in a Th2 response. Host microRNA responses (such as a decrease in miR-203b-3p) have been shown to indirectly stimulate JAK2/STAT3 signaling by increasing the levels of IL-6 through the regulation of the suppressor of cytokine signaling (SOCS3), which normally suppresses the downstream effects of IL-6. Red arrows depict secondary processes associated with the conceptual food reaction pathway.

**Table 1 genes-13-02024-t001:** Participant Characteristics.

Characteristic, *n* (%)	All (*n* = 164)	Food Allergy (*n* = 34)	No Food Allergy (*n* = 130)
**Maternal traits**			
Age, years; mean (SD)	30 (4)	32 (4)	30 (4)
BMI, kg/m^2^; mean (SD)	27.5 (6)	27.5 (6)	27.5 (6)
Maternal food allergies	14 (8.5)	8 (23.5) *	6 (4.6)
Maternal asthma	23 (14.0)	4 (11.7)	19 (14.6)
Maternal allergic rhinitis	36 (21.9)	9 (26.4)	27 (20.7)
Family food allergies in first- or second-degree relative	15 (9.1)	6 (17.6)	9 (6.9)
**Infant Traits**			
Female sex	96 (58.5)	22 (64.7)	74 (56.9)
Gestational age, weeks; mean (SD)	38.9 (1)	39.0 (1)	38.9 (1)
Race			
African American	7 (4.3)	0 (0.0)	7 (5.4)
Asian	7 (4.3)	3 (8.8)	4 (3.1)
Bi-racial	13 (7.9)	2 (5.9)	11 (8.5)
Other	11 (6.7)	1 (2.9)	10 (7.7)
White	126 (76.8)	28 (82.4)	98 (75.4)
Birth weight, g; mean (SD)	3352 (447)	3373 (416)	3341 (455)
**Environmental Exposures**			
Vaginal delivery	132 (80.5)	26 (76.4)	106 (81.5)
Earliest formula introduction			
None in initial 24 weeks	67 (43.8)	11 (34.4)	56 (46.3)
16–24 weeks	16 (10.5)	5 (15.6)	11 (9.1)
4–16 weeks	15 (9.8)	1 (3.1)	14 (11.6)
0–4 weeks	55 (35.9)	15 (46.9)	40 (33.1)
Solid food introduced by 4 months	22 (13.4)	5 (14.7)	17 (13.0)
Solid food introduced by 6 months	101 (80.8)	19 (55.8)	82 (63.0)
Maternal tobacco use	22 (13.4)	3 (8.8)	19 (14.6)
People in household, (range)	4 (2–9)	4 (2–9)	4 (2–9)
**Salivary factors**			
Time of collection, 24 h; mean (SD)	12 (3)	12 (3)	11 (3)

* *p* < 0.05. Note that parental estimates of earliest formula introduction were available for only 153/164 participants and estimates of solid food introduction at 6 months were available for only 125/164 participants. Abbreviations: Body mass index (BMI).

## Data Availability

We plan to deposit FASTQ files from RNA sequencing into the NIH GEO database upon acceptance of the manuscript.

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
