# Peer review of "Multi-Omic Profiles in Infants at Risk for Food Reactions"

_genes, 2022, doi:10.3390/genes13112024_

Round 1

Reviewer 1 Report

Few comments that require author's attention. These suggestions will help to improve the manuscript's quality. Please revise carefully. 

Please add a conclusion section. 

Figure 1 Please make the figure clear to read 

Figure 2 and Figure 3 Number each small figure, such as 2a, 2b, 2c... 

Author Response

We thank the reviewers for contributing their time and expertise to review our manuscript. We appreciate their constructive feedback and have done our best to incorporate all of their recommendations into the revised paper. Below is a point-by-point response to each reviewer comment:

Reviewer: 1

Comments:

Please add a conclusion section.

Thank you for this suggestion. We have added conclusion section, summarizing the most pertinent findings and their potential clinical implications for children with food reactions.

Figure 1 Please make the figure clear to read

We apologize for the lack of clarity. We have increased the size and resolution of Figure in the revised manuscript.

Figure 2 and Figure 3 Number each small figure, such as 2a, 2b, 2c

We apologize for this oversight. We have numbered each small figure and corrected the figure legends to reflect the numbering. 

Reviewer 2 Report

Dear Authors,

the manuscript is interesting, well-written and well-presented.

The topic addressed about infants allergies is of interest, because nowadays impactful and spread, in particular with regard to the multi-omic approach used in this cohort study.  Study design is valid and results are clear and promising for a future clinical application.

I suggest to implement the introduction with a brief mention about immuno-enzymatic and molecular assays (recent references) used for allergen detection in foods: a clinical survey in patients have to go hand in hand with a valid tools for monitoring of labels, so the allergens presence (also in traces) in foods (Reg. UE 2021/382; Reg. UE 1169/2011).

Furthermore, also in material and methods section I suggest to add a paragraph about: i. past family allergies; ii. maternal allergies; iii. type of allergy, both more common and investigated in the study.

Have you considered Vitamin D/probiotics/prebiotics supplementation in mother diet during pregnancy? Several studies are in progress about thier potential preventive and beneficial role also in FA (Murdaca et al. 2021).

Please, better check the reported references also in accordance with the format required by “Genes-MDPI”.

Best regards,

M.T

Author Response

We thank the reviewers for contributing their time and expertise to review our manuscript. We appreciate their constructive feedback and have done our best to incorporate all of their recommendations into the revised paper. Below is a point-by-point response to each reviewer comment:

Reviewer: 2

COMMENTS: The manuscript is interesting, well-written and well-presented. The topic addressed about infants allergies is of interest, because nowadays impactful and spread, in particular with regard to the multi-omic approach used in this cohort study.  Study design is valid and results are clear and promising for a future clinical application.

I suggest to implement the introduction with a brief mention about immuno-enzymatic and molecular assays (recent references) used for allergen detection in foods: a clinical survey in patients have to go hand in hand with a valid tool for monitoring of labels, so the allergens presence (also in traces) in foods (Reg. UE 2021/382; Reg. UE 1169/2011).

Thank you for this important suggestion. We agree that understanding and detecting allergens is crucial to understand the origin of food reactions. The revised manuscript now includes information on reliable methods for identifying allergenic ingredients in food products, including protein and DNA/RNA detection through Enzyme-linked immunosorbent assay, real time PCR and DNA biosensors (Linacero R et al. 2022).

Furthermore, also in material and methods section I suggest to add a paragraph about: i. past family allergies; ii. maternal allergies; iii. type of allergy, both more common and investigated in the study.

We appreciate the suggestion to include this important information. The revised manuscript now contains information regarding family food allergy history, maternal food allergy history, and other common allergic conditions in mothers (i.e., asthma and allergic rhinitis). This data has been added to the methods, the results, and Table 1. Unfortunately, we did not collect information about the specific food allergens for infant family members.

Have you considered Vitamin D/probiotics/prebiotics supplementation in mother diet during pregnancy? Several studies are in progress about their potential preventive and beneficial role also in FA (Murdaca et al. 2021).

Thank you for this interesting suggestion. We have added a paragraph to the introduction summarizing the potential interaction between vitamin D, host immunity, and microbial composition from Mudaca and colleagues (2021). Unfortunately, we did not collect data from our cohort regarding maternal use of these supplements, so we cannot assess their potential impact on microbial-host interactions in this paper. It will be interesting to see if interventional studies that provide these supplements detect changes in the molecular network we describe in our infant cohort.  

Please, better check the reported references also in accordance with the format required by “Genes-MDPI”.

Thank you for bringing this to our attention. We have closely reviewed the references and edited them to reflect the required format by “Genes-MDPI.”